# Cold Atmospheric Plasma Cancer Treatment, a Critical Review

**Dayun Yan** [1],*, **Alisa Malyavko** [2], **Qihui Wang** [1], **Li Lin** [1], **Jonathan H. Sherman** [3] **and Michael Keidar** [1],*

1 Department of Mechanical and Aerospace Engineering, George Washington University, Washington, DC 20052, USA; qwang52@gwmail.gwu.edu (Q.W.); lilin@email.gwu.edu (L.L.)

2 School of Medicine and Health Science, George Washington University, Washington, DC 20052, USA; alisamalyavko@gwmail.gwu.edu

3 WVU Medicine-Berkeley Medical Center, West Virginia University, Martinsburg, WV 25401, USA; jsherman0620@gmail.com

* Correspondence: ydy2012@gwmail.gwu.edu (D.Y.); keidar@gwu.edu (M.K.)

**Abstract:** Cold atmospheric plasma (CAP) is an ionized gas, the product of a non-equilibrium discharge at atmospheric conditions. Both chemical and physical factors in CAP have been demonstrated to have unique biological impacts in cancer treatment. From a chemical-based perspective, the anti-cancer efficacy is determined by the cellular sensitivity to reactive species. CAP may also be used as a powerful anti-cancer modality based on its physical factors, mainly EM emission. Here, we delve into three CAP cancer treatment approaches, chemically based direct/indirect treatment and physical-based treatment by discussing their basic principles, features, advantages, and drawbacks. This review does not focus on the molecular mechanisms, which have been widely introduced in previous reviews. Based on these approaches and novel adaptive plasma concepts, we discuss the potential clinical application of CAP cancer treatment using a critical evaluation and forward-looking perspectives.

**Keywords:** cold atmospheric plasma; reactive species; cancer treatment; physical treatment

## 1. Introduction

CAP is a near room temperature ionized gas, generated under non-equilibrium conditions [1–4]. The chemical reaction that occurs during the discharge process is complex and has been systematically introduced in previous reviews [5–7]. Here, we introduce the basic picture of CAP to facilitate better understanding of three treatment approaches. One schematic description of bulk CAP and the formation of reactive species in solution is shown in Figure 1a. Chemical composition in gas phase is quite different from that in liquid phase. Several reviews and studies have discussed the matter transition from gas phase into aqueous solutions [8–12].

Three types of CAP sources have been widely used in plasma medicine laboratories and clinical application. These include volume dielectric barrier discharge (DBD) source, surface discharge source, and CAP jet source. One typical source is based on the discharge that occurs between an electrode connecting to high voltage power and a grounded sample (Figure 1b). The electrode is coated with a dielectric material layer to avoid the formation of an equilibrium discharge, such as an arc discharge [13,14]. For a surface discharge source, discharge occurs between the grounded electrode and the high voltage electrode coated with a dielectric layer (Figure 1c). Bulk plasma tends to be generated in the gap between electrodes. The gaps between electrodes can be designed to form complex structures and patterns, which are widely used in wearable CAP devices [15–18]. The CAP jet is another typical source (Figure 1d). In this case, discharge occurs between electrodes. The sample is not involved in the discharge process. A dielectric material covers at least one electrode [19–22]. A noble gas, such as helium (He) or argon (Ar), is used as the carrying gas to trigger discharge at a relatively low discharge voltage and transport the ionized gas out of a nozzle [23,24]. Some gases, such as nitrogen ($N_2$) and oxygen ($O_2$), can also mix with noble gas to modulate chemical components of CAP [22].

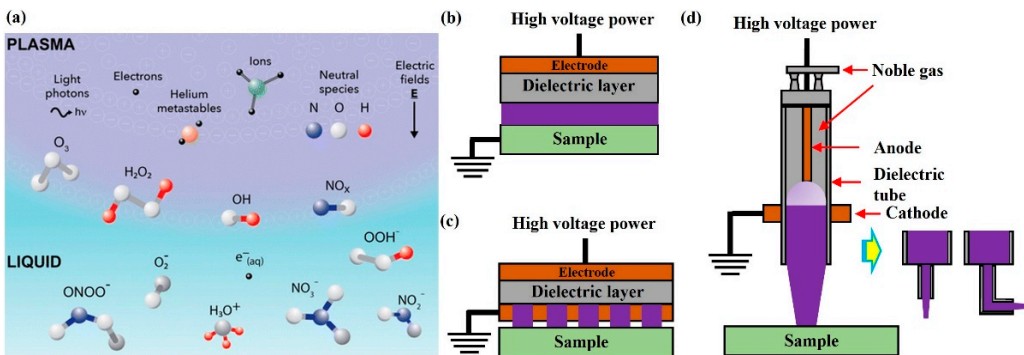

**Figure 1.** CAP sources used in plasma medicine. (**a**) A schematic illustration of typical component of CAP and reactive species formed in the CAP-treated liquid (reproduced with permission from Tumor Biology 37.6, 7021–7031 (2016). Copyright 2016 Springer Nature). (**b**) A volume DBD source. (**c**) A surface discharge source. (**d**) A CAP jet source. Volume and flow rate of jet can be modulated by the nozzle.

## 2. Treatment Based on CAP-Activated Solutions

Conventional CAP-based cancer treatment is performed in a chemically based manner, including indirect CAP treatment and direct CAP treatment strategies. The basic principle of chemical-based indirect CAP treatment is using CAP-treated or -activated solution (PAS) or medium (PAM) to affect the growth of cancer cells via reactive species and other possible plasma-generated products in an aqueous solution [25]. Such an approach is not novel. The use of CAP-treated water or underwater discharge is a long-standing approach, and leads to the death of bacteria or other microorganisms due to the cytotoxicity of reactive species [10,26–32]. If these CAP-originated chemical factors are toxic to microorganisms, it is not fully unexpected that these can also be toxic to cancer cells. A more attractive feature of PAS/PAM is the selective anti-cancer performance in many cell lines [33–35]. However, this selectivity is solely determined by the sensitivity of cancer cells or normal cells to long-lived reactive species or other formed components.

PAS/ PAM can be derived using a CAP treatment above the surface of the solution (Figure 2). Contact between the bulk plasma and the medium is necessary. Several long-lived reactive species, such as $H_2O_2$, $NO_2^-$, $NO_3^-$, and ONOO−, have been observed in the aqueous solutions after treatment [36,37]. The reaction products of CAP and original components also form in other solutions, such as lactate solution [38]. In terms of cell culture medium, CAP will result in complex reactions between ROS/RNS and medium components, particularly amino acids, such as cysteine, methionine, and tryptophan [39,40]. PAS can also be derived using the discharge in solution [9,34].

The largest advantage of PAS is thought to be its ability to serve independently from the CAP source once PAS has been derived [39,41–43]; that is, indirect treatment can be used as a pharmacological modality. However, due to its pharmacological nature, a clear understanding of its chemical processes and mechanisms needs to be obtained before it can be accepted by the market. To date, only some long-lived reactive species, such as $H_2O_2$, $NO_2^-$, $NO_3^-$, and ONOO−, in addition to a small number of other components in PAS, have been regarded as being effective anti-cancer components [39,42,44]. In general, a single chemical factor cannot explain the observed cytotoxicity. This is demonstrated by a synergistic use of $H_2O_2/NO_2^-$, or $H_2O_2/NO_2^-/NO_3^-$. Varying ratios of these chemical combinations can achieve similar results to those seen after an indirect CAP treatment [45–47]. To date, the uniqueness of indirect treatment compared with a simple long-lived reactive species treatment has not been clearly explained or distinguished.

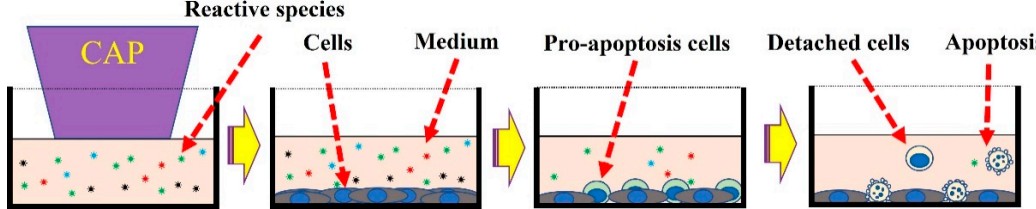

**Figure 2.** Schematic illustration of the indirect CAP treatment based on CAP-activated medium in vitro. Here, the feature of apoptosis is used to represent cell death.

An additional concern regarding practical implementation is the stability of PAS/PAM during storage. As a pharmacological modality, it must be stably stored under conditions which have relatively low-cost requirements. The extremely cold temperature ($-196\ ^\circ$C) of liquid nitrogen is an ideal storage condition, but this storage strategy has a high cost and is not practical when a large amount of PAM or PAS is needed. The ideal storage temperature should be either 2–8 $^\circ$C, $-20\ ^\circ$C, or room temperature. In many studies, however, PAM must be stored at -80 $^\circ$C or $-150\ ^\circ$C to realize full stability [38,42]. PAM experiences degradation at room temperature and at $-20\ ^\circ$C [42,48]. This degradation is slightly inhibited at 2–8 $^\circ$C [42,48]. The natural degradation of PAM throughout a wide temperature range may severely limit its potential pharmacological value.

The degradation mechanism provides clues to solve this dilemma. The degradation of PAM is mainly due to the reaction between amino acids, particularly cysteine and methionine, and reactive species [49]. If the solution or medium does not contain cysteine or methionine, the stability of PAM during storage drastically improves, even at 2–8 $^\circ$C [49]. Furthermore, the discovery of 3-nitro-L-tyrosine as an anti-degradation additive also provides another approach to resist the degradation of PAM at 2–8 $^\circ$C, even with the presence of cysteine and methionine [49]. For a simple solution, such as phosphate-buffered saline (PBS), 3-nitro-L-tyrosine can fully retain $H_2O_2$ in CAP-treated PBS for 7 days at 8 $^\circ$C [50].

## 3. Chemical-Based Direct CAP Treatment

The earliest attempts to test the anti-cancer capacity of CAP treatment were conducted using direct treatment. Direct treatment is also the most conventional approach [51]. It is based on a simple assumption that a treatment should be performed when cells are in culture conditions or at least in an environment close to that of the standard culture conditions. A layer of medium covered cells during most studies involving direct CAP treatment [52]. Because thermal irradiation, UV irradiation, and EM emission may be blocked by this layer of medium, particularly when this layer is adequately thick, the physical effect has not been noted in most previous studies. This medium layer also plays the role of an interface between the gas phase of CAP and aqueous solutions [8]. Without such an aqueous environment, many reactions may be interfered with or fully blocked.

Direct and indirect treatments share many similarities. Nearly all cellular damage can be observed both in direct and indirect treatment. Common cellular responses to direct treatment include the rise of intracellular ROS, DNA damage, mitochondrial damage, endoplasmic reticulum damage, and damage to the cytoplasmic membrane [42,53–57]. These cellular changes have also been observed during indirect treatment [33,34,42,58,59]. Sensitivity of cancer cell lines to direct treatment and indirect treatment is also usually similar. The unique sensitivity of cancer cell lines to chemical-based treatment may be partially due to the specific anti-oxidant system in different cancer cell lines, which can be related to differential expression level of TP53 in various cell lines [60,61]. Some cancer cell lines, such as melanoma cell line B16F10, are very resistant to reactive species in both direct and indirect treatments [62]. In a case like this, a long period of CAP treatment is necessary. This is a natural limitation of chemical-based CAP treatment.

Apoptosis is the main cell death after chemical-based treatment; however, necrosis and autophagy-associated cell death have been also observed [56,57,63–67]. Apoptosis is

a slow process that begins with the formation of clear blebs, representing apoptotic cells losing their pseudopodia. Cells may later detach from the substrate or may experience apoptosis on the substrate. It should be noted that cancer cells may recover to their normal shape depending on the treatment dose. Bubbling can also be observed in some cells during the cell death process, which is believed to be a feature of another form of necrotic cell death known as pyroptosis [68].

One aim of plasma medicine may be the exploration of unique cellular responses in the CAP-treated cancer cells, rather than repetitively showing the features of other methods, such as apoptosis and cell cycle arrest [69]. Quantitative comparison between direct and indirect treatment provides clues to understand the unique features of direct CAP treatment. In the studies comparing the cytotoxicity of direct and indirect CAP treatment under the same experimental conditions, the former caused a stronger effect than the latter in many cases [70,71].

The unique features of direct treatment may contribute to these differences (Figure 3). One such feature is the activation phenomenon of CAP-treated cancer cells. The treated cancer cells quickly enter an activated state in which they become very sensitive to the cytotoxicity of reactive species, particularly $H_2O_2$ [72]. In contrast, cancer cells do not enter an activation state after indirect treatment or simple long-lived reactive species treatment. Therefore, cell activation following direct treatment may be due to short-lived reactive species or physical factors. The activated cancer cells remain in this unique state for approximately 5 h [72]. Subsequently, cancer cells become completely desensitized. The mechanism of activation is still unknown. Activation can also be named sensitization. One of our studies demonstrated that the activated U87MG cells can be sensitized to the cytotoxicity of a typical chemotherapy drug, temozolomide (TMZ) [73]. Another recent study confirmed that just using the EM emission from the ionized helium in a sealed discharge tube can also drastically sensitize glioblastoma cell lines U87MG and A172 to the cytotoxicity of TMZ without causing a noticeable effect on the normal astrocyte cell line hTERT/E6/E7 [74]. Thus, the activation or sensitization of the CAP-treated cancer cells to reactive species or chemotherapy drugs may become a new direction in plasma medicine, i.e., not to directly to kill cells, but to sensitize cancer cells to traditional chemotherapy by either chemical factors or physical factors.

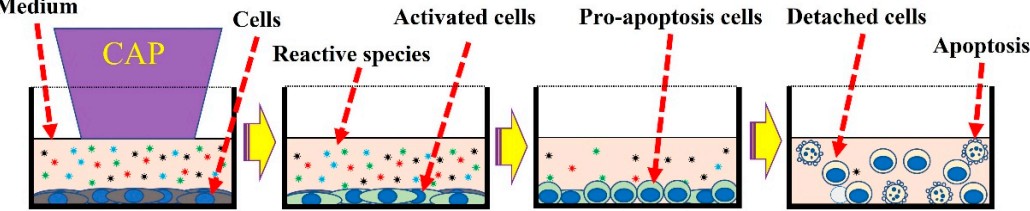

**Figure 3.** Schematic illustration of a typical direct CAP treatment in vitro. Here, the feature of apoptosis is used to represent the cell death pathway. Compared with indirect CAP treatment, cell-based $H_2O_2$ generation and activation phenomenon in direct CAP treatment drastically enhances the cytotoxicity on cancer cells when experimental conditions are the same.

Another unique cellular response to direct treatment is the strong cell-based $H_2O_2$ generation at a micromolar level. For many cancer cell lines, the concentration of extracellular $H_2O_2$ in a medium with cells immediately after direct treatment is significantly higher than in a case in which solely medium is exposed to treatment without cells [70,75,76]. Clearly, the additional $H_2O_2$ generation should be due to the presence of cancer cells. The larger the discharge voltage of CAP, the greater the cell-based $H_2O_2$ generation [77]. Short-lived reactive species in CAP, such as superoxide, may trigger a dismutation reaction catalyzed by extracellular superoxide dismutase (Ex-SOD) on the cytoplasmic membrane.

## 4. Treatment Based on Physical Factors

To overcome the sensitivity limitation of cancer cells to reactive species, physical-based CAP treatment was recently proposed. It is different from previous approaches, not only because of the stronger anti-cancer effect in some cell lines, but also because of the novel underlying mechanism. Even this concept has not been widely accepted in academia, and the preliminary discussion relates to current in vitro studies involving six cancer cell lines. To demonstrate the physical-based biological effect, we harnessed a novel strategy. We treated the bottom of an inverted cell culture dish or multi-well plate rather than using the typical treatment approach of directly exposing cells or solutions to bulk CAP. The treated cells were immediately moved back into standard culture conditions by replenishing medium. This strategy blocked all chemical factors due to the presence of polyene material of cell culture dishes and multi-well plates, with a thickness of approximately 1 mm (Figure 4). Neither UV nor thermal irradiation caused any observable cellular change [78,79]. EM emission generated by CAP caused the observable cellular changes. Physical-based treatment is not necessary to cause stronger cytotoxicity in cancer cells compared with chemical-based treatment. To date, six cell lines have been investigated: glioblastoma cell line U87MG, melanoma cell line B16F10, breast cancer cell line MDA-MB-231, lung cancer cell line A549, bladder cancer cell line MB49, and MBT2 [78–80]. Among these, nearly all cell lines show a stronger sensitivity to physical-based treatment, with the exception of the bladder cancer cell line MT49 [80].

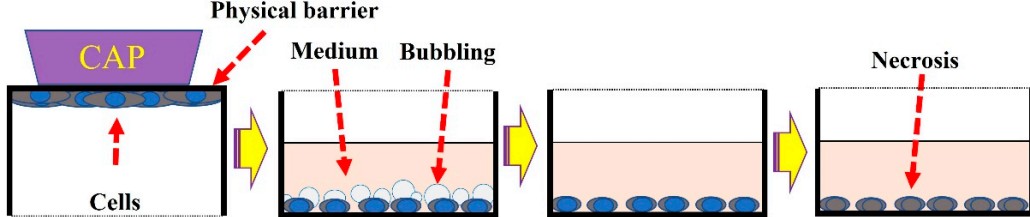

**Figure 4.** Schematic illustration of a physical-based CAP treatment in vitro. Here, the feature of necrosis is used to represent the cell death pathway observed in references. Quick bubbling and cytosolic shrinkage are two key features of physical-based cell death.

The cell death after physical-based treatment is clearly a form of necrosis, rather than apoptosis. Cellular changes can be observed approximately 1–2 min after treatment, and are characterized by a fast shrinkage of cytosol toward the nucleus and rapid leakage of cellular solutions forming bubbles on cellular surface. Bubbling can be observed on nearly all treated cells and the growth of bubbles takes place 8–10 min after treatment, after which the growth of bubbles ceases. Bubbles remain on the cellular membrane for about 1–2 h, with most of them detaching after this time. The basic cellular shape does not change for days after bubble detachment. However, DNA in the nucleus is not seen two days after treatment. Additionally, cells do not display normal cellular functions, such as duplication, movement, and apoptotic features, after treatment. It appears that cells are fixed after physical-based treatment.

Unlike traditional chemical-based treatment, the efficacy of physical-based treatment may be influenced by the power of EM emission and the spatial position between CAP source and target. Physical factors can affect a significantly larger area than the contacted area of bulk CAP on the target, and even these areas are physically separated; for example, in a 96 well plate, the tip size of the CAP jet is roughly close to the diameter of a single well of the plate. At a minimum, the 10 wells close to the treated single well also experience strong growth inhibition due to physical-triggered cell death [78,79]. These wells were physically separated by the walls of well. In contrast, the same chemical-based treatment only causes damage in the single CAP-treated well. In contrast, however, physically triggered cellular response can only be observed when the CAP source and target have a suitable gap. The direct contact of bulk CAP with the surface of the barrier (bottom of plate

or dish) does not always trigger cell death. Surprisingly, when the detachment of bulk plasma from the target is set to a small range, such as 8 mm, a strong physically triggered cell death can still be triggered [78]. The transmission of EM waves from the tip of bulk CAP may be responsible for these two features.

Physical factors in CAP not only affect cancer cells in physical-based treatment but may also impact cells during direct treatment. As we mentioned above, a chemical effect during direct CAP treatment is inevitable for the directly exposed cells in vitro, even without the coverage of medium. Thus, both physical and chemical factors coexist in this direct treatment. The volume of the medium layer determines the factors that play the dominant role [78].

The trans-barrier capacity of physical-based CAP treatment may facilitate the non-invasive application of CAP in medicine. Plastic materials, such as polystyrene and UV/heating reflection films made of complex dielectric materials, cannot inhibit the penetration of physical factors [78,79]. In contrast, any of these barriers can completely block all chemical factors. Conductive materials can block the transmission of EM emission from bulk CAP. In the example of a 96 well plate, setting a copper sheet (ungrounded) between the tip of the CAP and the bottom of the plate could completely block physical-triggered cell death if the size of the copper sheet is adequately large, such as covering an area of $5 \times 5$ wells. If the copper sheet is only sufficiently large to cover a small area, such as just directly covering the well touched by the bulk CAP, physical-triggered cell death may not be inhibited [78,79]. Recently, the sensitization of glioblastoma cell lines to chemotherapy drugs by the EM emission from a helium discharge tube further demonstrated the contactless and transbarrier potential of physical factors in CAP treatment.

## 5. Anti-Cancer Capability In Vivo

The final goal of plasma medicine is its clinical application. At present, the pathway to realize this goal remains an open question. Three treatment approaches in vitro have provided some projections for potential clinical fates. In terms of animal studies, there are three main models: subcutaneous, intraperitoneal, and orthotopic tumor models [38,81,82]. In some cases, intracranial models have also been presented, in which PAM/PAS has been injected to affect the growth of tumors [38,58,83,84]. In most cases, an anti-cancer effect was demonstrated on the subcutaneous tumor model. It should be noted that, for subcutaneous animal models, tumors are never directly exposed to bulk CAP unless surgery is performed prior to treatment to gain access to it. This is a fundamental difference between in vivo and in vitro studies. In the former case, the skin and subcutaneous tissues face the impact of bulk CAP before tumor tissue. If reactive species, particularly ROS such as $H_2O_2$, cannot penetrate the barriers above the tumor tissue, the conclusions based on direct CAP treatment in vitro may not be very useful to explain in vivo treatment.

In contrast, physical-based treatment may build a reasonable correlation between in vitro and in vivo studies. Because of a general conclusion that reactive species cause all observable effects, the potential roles of the EM effect and even the heating effect have not been carefully explored, particularly for the cases in which the authors used plasma with a significantly higher temperature than room temperature. Strictly speaking, these plasmas are not cold plasma. In short, it is still unknown whether physical factors play a dominant role during the studies involving direction contact between plasma and tissues/cells.

Many animal studies have been performed over the past decade, and the results of these studies provide hope for the use of CAP as a clinic anti-cancer modality. The earliest animal study used a glioblastoma U87MG xenograft tumor model to test the anti-cancer effect in mice [81]. This study was the first to demonstrate that treatment above the skin of a subcutaneous tumor can achieve a promising half decrease in tumor size [81]. A fractionated treatment resulted in a greater effect than a single, longer plasma treatment [85]. Several subsequent studies demonstrated a similar strong anti-cancer effect during treatment in subcutaneous tumor models, including bladder cancer, melanoma, pancreatic carcinoma, glioblastoma, neuroblastoma, head and neck cancer, and a breast

cancer xenograft model [54,82,86–91]. The safety of CAP treatment in animals has been proved in some studies, which confirmed no thermal damage or other obvious damage occurred to the surrounding normal tissues after treatment [86].

Due to the natural barrier of skin and other tissues above tumorous tissues, several attempts have been made to explain whether an inhibition effect is present during the transdermal treatment in subcutaneous tumor models. If we do not consider the potential anti-cancer effect due to physical factors, CAP-originated reactive species may need to penetrate skin and other tissues above tumorous tissues [92]. Moreover, reactive species such as ROS may just penetrate subcutaneous tissue with a limited depth but act as a second messenger to trigger other biological effect. To date, a small number of studies suggest the possible transdermal diffusion of reactive species across some skin substitutes, such as agarose gel [93–96]. However, this conclusion requires more experimentation and evidence to be fully supported. The CAP-triggered immune response is a promising candidate to explain the anti-cancer effect in vivo [97]. CAP may activate the immune system to attack tumorous tissue. Such immune responses have been referred to as induction of immunogenic cancer death (ICD) [98,99]. CAP treatment performed on tumor tissue may trigger cancer cells to emit signals, such as damage-associated molecular patterns (DAMPs), which can attract and stimulate local immune cells [98,99]. Typical DAMPs include calreticulin (CRT) and adenosine triphosphate (ATP). In another study, the production of inflammatory cytokines (IFN-γ) from the splenocytes of CAP-treated mice was noted [100]. If we consider the possible transdermal physical-triggered cell death discussed above, the noticeable leakage of cellular solutions immediately after treatment may trigger an inflammatory response or immune response in vivo. This explanation does not depend on the prerequisite criteria of reactive species penetrating the skin. In this scenario, reactive species may not be necessary for in vivo anti-cancer effect.

## 6. An Outlook of CAP Cancer Treatment

Recently, clinical tests have been performed by various organizations in which CAP treatment was incorporated into patient treatment plans. US Medical Innovation, LLC performed a clinical test where a CAP jet was used to treat stage 4 colon cancer remnants at Baton Rouge General Medical Center in Baton Rouge, Louisiana, after initial surgery to remove a tumor [101]. Another test in Germany was performed by treating six patients with a locally advanced (pT4) squamous cell carcinoma of the oropharynx, and who were suffering from open infected ulcerations [102]. CAP treatment noticeably inhibited the growth of tumors in some patients and caused a reduction in odor and the use of pain medication [102]. Partial remission was achieved for at least 9 months in two patients after CAP treatment. Incisional biopsies found a moderate amount of apoptotic tumor cells and a desmoplastic reaction of connective tissue. CAP has also been used to treat oral cancer with very mild side effects, such as a bad taste during and after treatment and minimal discomfort during treatment. None of these side effects were life threatening and most were transient with full recovery of normalcy [103].

These clinical tests have a common feature: bulk CAP directly touches tumorous tissues without the cover of a bulk medium or an aqueous layer with a macroscale. Such a working condition is essentially different from the experimental conditions that are used in in vitro studies. Due to the absence of a bulk aqueous layer, it is highly probable that the physical factors in CAP play an important role in clinical application.

Based on the discussion above, the following predictions for possible approaches of CAP-based cancer treatment can be made (Figure 5). For subcutaneous tumors, a direct treatment strategy just above the skin of the tumor site is recommended. Based on the observation from physical-based CAP treatment in vitro, we speculate that even an air gap, such as several mm between bulk CAP and the skin, will not fully weaken the treatment's efficacy. In contrast, the efficacy of treatment may be fully inhibited when bulk CAP completely touches the skin too closely. For intraperitoneal tumor or tumors residing in deeper tissues, simply treating the skin is a questionable approach based on the current

evidence and conclusions. If considering the anti-cancer capacity due to physical factors, the effective penetration depth remains unknown. PAS or PAM may be used to treat the tumor in such cases by administering subcutaneous or intraperitoneal injections. It is important to mention that CAP may be synergistically used with endoscopic technology to treat deep tissue tumors by a minimally invasive surgery [104]. In addition to these cases, CAP can also be used as an adjunct to current surgical methods. Bulk CAP can come in direct contact with the exposed tumor tissue or the remnants of tumor tissue during surgery. This direction may be another promising candidate.

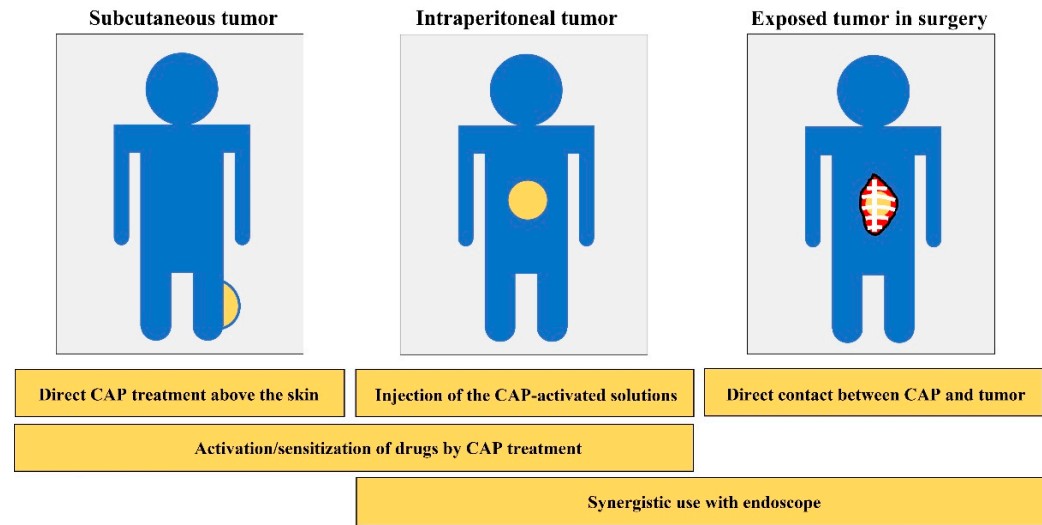

**Figure 5.** Schematic illustration of possible means to use CAP in clinical cancer therapy.

One unique feature of CAP is its ability to quickly change its composition and key parameters on demand and subject to specific requirements [105]. An illustration of the adaptive CAP concept is presented in Figure 6a. This concept is based on the capacity to read the cellular response to CAP in real time and to modify plasma composition and power in real time via a feedback mechanism [106,107]. Direct treatment is the prerequisite to realize such adaptive plasma sources. Adaptive CAP enables the modification of plasma conditions in real time to optimize the effect on cancer and normal cells. An adaptive plasma therapeutic system can adjust its interaction with cells and tissues by responding to the boundary conditions at the cold plasma–cell interface [108]. Furthermore, due to the different effects of CAP on cancer cells and normal cells, an adaptive CAP system can achieve an enhanced selective anti-cancer effect by controlling the dose of reactive species or other possible interactions between cells and CAP. Figure 6b schematically illustrates how plasma interaction with cells can lead to the transition between different discharge modes, which enable possible plasma self-adaptation. Adaptive plasma sources having different operation modes with bifurcation points should be developed.

To implement an adaptive plasma system, a multi-parametric feedback system based on cellular responses needs to be developed. One example is a platform demonstrated in Ref. [109], which monitors the cellular response to CAP treatment in a continuous read approach, using RealTime-Glo assay to achieve a feedback pathway [109]. The feedback algorithm relies on the cell viability measurements of cancer cells and normal cells to predict an optimized amount of plasma delivery power and treatment time. These two parameters regulate CAP dose and further determine the concentration of ROS/RNS delivered to cells during direct CAP treatment.

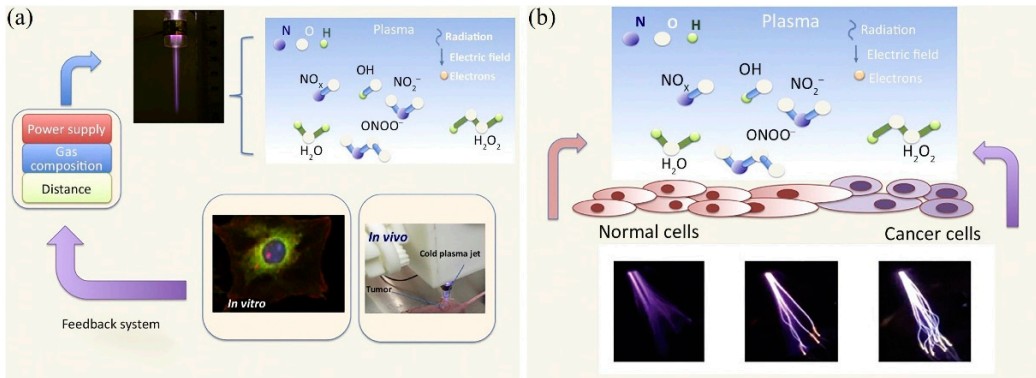

**Figure 6.** A schematic illustration of an adaptive plasma system. (**a**) Basic concept. A CAP treatment can trigger different responses in cancer cells and normal cells. An adaptive plasma system can collect, measure, record, and analyze these responses as signals. These signals provide a feedback signal to the system, which in turn modulates the chemical composition of CAP by controlling gas composition, power supply, the gap between the source and target, etc. (**b**) The target affects the discharge pattern of plasma by self-organization. For example, cancer cells and normal cells exhibit large differences in membrane potential, which can be used as a feedback signal to control plasma through the self-organization of plasma with different patterns and modes. The plasma adaptive therapeutic system through real-time cellular and tissue response can lead to optimal and selective cancer treatment ((**a**,**b**) are reproduced with permission from Keidar, Michael, et al. Trends in Biotechnology 36.6, 586–593, (2018). Copyright 2017 Elsevier Publisher. [107]).

A feedback control algorithm can be developed using a nonlinear model-predictive control (MPC). MPC or receding-horizon control is a strategy used to implement a numerical solution of an open-loop optimization in feedback. MPC is particularly desirable for an adaptive plasma system, because treatment conditions are determined in an optimal fashion to maximize anti-cancer efficacy. The algorithm has been shown to leverage a model of plasma tumor interactions [110]. Such a feedback system can employ adaptive plasma properties to yield optimal and appropriate treatments tailored to the real-time diagnostics [111].

It should be noted that such an adaptive approach can ultimately lead to a personalized CAP-based cancer treatment and, possibly, treatment of other diseases. Personalized medicine is currently being investigated in cancer therapy and an adaptive plasma system would fit well into this multi-disciplinary treatment approach. One goal of personalized medicine is to customize the treatment to the patient's specific needs based on their individual genetic makeup. Application of plasma may lead to unique individual-based responses. The identical combination of reactive species and electric field applied to the body during plasma treatment may have a different effect on each person due to genome specifics.

## 7. Summary

At present, three cancer treatment approaches exist based on CAP technology. Each presents its own advantages and drawbacks, particularly regarding the limitations in clinical application. Overcoming technical barriers of using CAP, identifying an optimized strategy to fully explore the anti-cancer potential of CAP technology, and implementing CAP into clinical cancer treatment, are complex challenges for plasma medicine that lay ahead. This perspective evaluation of the three current CAP cancer treatment approaches may provide inspiration to achieve the final goal of clinical application.

**Author Contributions:** D.Y. and M.K., methodology; D.Y., A.M., Q.W., L.L., J.H.S. and M.K., writing—review and editing. All authors have read and agreed to the published version of the manuscript.

**Funding:** This research was funded by National Science Foundation grant, grant number 1747760.

**Institutional Review Board Statement:** Not applicable.

**Informed Consent Statement:** Not applicable.

**Data Availability Statement:** Not applicable.

**Conflicts of Interest:** The authors declare no conflict of interest.

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
