# Peer review of "Cold Atmospheric Plasma Cancer Treatment, a Critical Review"

_applsci, doi:10.3390/app11167757_

Round 1

Reviewer 1 Report

The present review article on the application of physiological plasma deals with an important and novel topic in the field of oncological therapy.
However, compared to the numerous review articles on this topic in recent years, there are no new aspects in the manuscript. The direct and indirect treatment of cells has been extensively studied and the effects are well accepted. In general, a summary of current in vitro experiments is not necessary. This is different for the animal experiments and clinical studies performed with physical plasma. Here, more breadth would be desirable, but this is not implemented. The exotic topic of the physical effect of such a treatment, which is independent of chemical factors, is not established and is based on only a few (presumably own) publications. The time is not yet ripe to address this aspect in a review.
The topic is undoubtedly important, but the present manuscript is unfortunately not, since the contents have already been presented many times in very recent reviews. The overloaded and sometimes superfluously long presentation of some aspects cannot cover this. I therefore recommend the rejection of the manuscript.

Author Response

See response. 

Reviewer 2 Report

Dear Authors,

The manuscript applsci-1326280, entitled 'Cold Atmospheric Plasma Cancer Treatment, a Critical Review.' present the cold plasma sources for cancer treatment. This review is well written and has nice results presented. I propose that this manuscript be considered for publication to Applied Sciences journal as is. 

As a general remark, I would expect, given the expertise of the authors and the interesting field, a more complex (supported by more information / pages ), considering the type of manuscript: review.

Author Response

See response. 

Reviewer 3 Report

This paper is a critical review about the use of cold atmospheric plasma (CAP) in cancer treatment. The authors after a concise but enough abstract evaluate a group of items related with the use of CAP and CAP-activated solutions. The descriptions were deep enough and always with the critical point of view.

However the authors should made minor alterations:

Point 5, page 6: in this sections the authors should also refer the orthotropic models.

References: pag. 10, ref. 78 should be corrected, the DOI does not correspond to the title showed.

Author Response

See response. 

Round 2

Reviewer 1 Report

Of course, I understand that the authors consider the manuscript novel and important. However, my overall assessment remains, the review article shows no novelty in the field and does not meet the criteria of the journal . The minor rewrite has not changed this.